# Chemical characterization, antimicrobial, and antioxidant potentials of *Juniperus phoenicea* wood tar

**Sadia Tina**[1]*, **Oussama Khibech**[2], **Samy Iskandar**[1], **Mohamed Bouhrim**[3], **Mohammed Al-zharani**[4]*, **Fahd A. Nasr**[4], **Ashraf Ahmed Qurtam**[4], **Allal Challioui**[2], **Balouch Lhousaine**[1], **Meryem El Jemli**[1]

1 Laboratory of Research in Drug Discovery, University Mohamed VI of Sciences and Health, Casablanca, Morocco, 2 Mohammed Premier University, Faculty of sciences, Department of Chemistry, Laboratory of Applied and Environmental Chemistry (LCAE), Oujda, Morocco, 3 Laboratoires TBC, Laboratory of Pharmacology, Pharmacokinetics, and Clinical Pharmacy, Faculty of Pharmaceutical and Biological Sciences, Lille, France, 4 Biology Department, College of Science, Imam Mohammad Ibn Saud Islamic University (IMSIU), Riyadh, Saudi Arabia

* stina@um6ss.ma (ST); mmyalzahrani@imamu.edu.sa (MA-z)

## Abstract

Wood tar obtained from *Juniperus phoenicea L.* is traditionally used in Morocco, yet its chemical profile and biological activities remain insufficiently documented. In this work, *Juniperus phoenicea* wood tar (*JPWT*) was chemically characterized by GC-MS, qualitatively screened for major phytochemical classes, and evaluated for total polyphenols, tannins and flavonoids contents. Antioxidant activity was assessed using DPPH and FRAP assays, and antibacterial activity was screened by the disc diffusion method. GC-MS identified 17 compounds, with hydroxychavicol (46.3%), 2,3-dehydroferruginol (18.5%) and eugenol (7.3%) as major constituents. JPWT showed high levels of polyphenols (242.91±68.93 μg GAE/mg), tannins (110.59±11.31 μg TAE/mg) and flavonoids (26.74±0.52 μg QE/mg). JPWT exhibited antioxidant activity with an IC50 of 17.05±0.67 μg/mL in the DPPH assay and an EC50 of 75.81±2.02 μg/mL in the FRAP assay, confirming its antioxidant activity. Antibacterial screening revealed dose-dependent inhibition, with stronger effects against Gram-positive strains. Molecular docking suggested that hydroxychavicol and 2,3-dehydroferruginol may contribute to antibacterial activity through favorable binding to MRSA PBP2a and ESBL CTX-M-15. Overall, these findings support JPWT as a promising natural source of antimicrobial and antioxidant agents and warrant further isolation and in vivo investigations.

## 1. Introduction

The Mediterranean basin is a globally recognized biodiversity hotspot and a major reservoir of aromatic and medicinal plants estimated at approximately 25000–30000

**Data availability statement:** Data are available from the: https://github.com/khibech/Juniperus-phoenicea-.

**Funding:** This work was supported and funded by the Deanship of Scientific Research at Imam Mohammad Ibn Saud Islamic University (IMSIU) (grant number IMSIU-DDRSP2601). The funders had no role in study design, data collection and analysis, decision to publish, or preparation of the manuscript.

**Competing interests:** No authors have competing interests.

species and subspecies [1]. It is in this large biodiversity hot-spot that the Iberian Peninsula and Morocco act as one of the two principal centers of plant diversity, with Turkey and Greece being the other [2]. With its Mediterranean climate, vast plains, and high mountain ranges rising to 4000 meters, diverse flora and variety are favorable for growing in Morocco. In recent years, natural products, especially medicinal plants, have attracted the scientific community as promising alternatives for treating various infectious diseases [3,4]. In this context, the antimicrobial and antioxidant properties of essential oils are well documented, leading to their application in pharmacology, medicine, microbiology, plant pathology, and food preservation [5,6]. However, the increasing emergence of multidrug-resistant microorganisms poses a serious global health challenge. The excessive use of antibiotics has significantly diminished their effectiveness, as demonstrated by the resistance observed in many bacteria notably *Escherichia coli* and *Klebsiella pneumoniae* [7]. Similarly, oxidative stress has emerged as a critical causative factor in aging and several chronic diseases such as diabetes, cancer, atherosclerosis, arthritis, and neurodegenerative disorders [8], All of which are often characterized by attendant inflammatory responses [9]. In response, plants produce variety of secondary metabolites including alkaloids, terpenoids, and phenolic compounds such as phenolic acids, flavonoids, tannins, lignins, quinones, and coumarins which are responsible for their antimicrobial, antioxidant, anti-inflammatory, and anticarcinogenic activities [10]. *Juniperus phoenicea L. (JP)* is a monoecious or dioecious coniferous tree or shrub belonging to the Cupressaceae family, characterized by its scaly leaves [11]. It plays a significant ecological role in the western Mediterranean basin, being considered a keystone species in cold-adapted open woodlands with steppe-like vegetation [12]. In addition to its ecological importance, *JP* has long been valued in traditional medicine for its hypoglycaemic, diuretic, antidiarrheal, antirheumatic, and broncho-pulmonary applications [13]. Among its derivatives, *Juniperus phoenicea* wood tar (*JPWT*), obtained by the destructive distillation of *JP* wood, is traditionally used for skin ailments and infections [14]. Such extraction methods remain largely unvalorized, with only a few studies devoted to them.

The present study aimed to qualitatively screen *JPWT* for major classes of secondary metabolites, to quantify total polyphenols, tannins and flavonoids, evaluate antioxidant activity by using DPPH and FRAP assays, evaluate antibacterial activity against selected Gram-positive and Gram-negative strains, and to use molecular docking as a complementary approach to explore potential molecular targets and provide mechanistic insights into the observed antibacterial activity.

## 2. Materials and methods

### 2.1. Plant materials

Wood of *Juniperus phoenicea L.* was collected from Termilat (Ifrane Province, Morocco; 33°30'02.1"N, 5°05'28.0"W). The plant material was taxonomically identified by Prof. Taleb Sghir. A voucher specimen (RAB114890) was deposited at the Herbarium of the Scientific Institute of Rabat (Morocco). Collected branches were examined for integrity and absence of dust and insect contamination. No specific permits or official authorizations were required for the collection of plant material, as

the species is not protected and the sampling sites are not located in restricted areas. Plant collection was carried out in accordance with local practices and regulations. Particular care was taken to minimize environmental impact and preserve the surrounding ecosystem.

## 2.2. Preparation of plant extracts

The wood (100 g) is placed in a flask (250 mL) and heated to carbonization (1 h). The steam released during distillation is conducted through a pipe and condensed in a cold settling tank. One hour later, two distinct layers are formed: an upper layer, made up of a brownish-yellow aqueous liquid, corresponding to the phase not used in traditional Moroccan medicine, and a lower layer, corresponding to the tar, which is collected in an opaque glass bottle.

## 2.3. Qualitative phytochemical tests

*JPWT* was qualitatively screened for major classes of secondary metabolites (alkaloids, tannins, polyphenols, flavonoids and saponins) using standard colorimetric tests [15,16].

**2.3.1. Alkaloids.** A total of 0.5 g of *JPWT* was mixed with 5 mL of 60% ethanol and then divided into two equal portions. The presence of alkaloids was confirmed by the formation of a reddish-brown precipitate after the addition of a few drops of Dragendorff's reagent, or by the appearance of a white precipitate following the addition of Mayer's reagent [15].

**2.3.2. Test for tannins.** The addition of a few drops of a 5% ferric chloride solution to a mixture of the *JPWT* extract and 2 ml of ethanol showed the appearance of blue coloration, indicating the presence of tannins [15].

**2.3.3. Saponins.** After adding 2 mL of distilled water to 2 mL of the ethanolic *JPWT* solution, the mixture was shaken for 1 minute. The formation of a persistent foam layer of approximately 1 cm after 15 min indicates the presence of saponins [17].

**2.3.4. Test for flavonoids.** When 1 ml NaOH was mixed with 2 ml ethanolic extract of *JPWT*, the presence of flavonoid was revealed by the appearance of a yellow color [18].

**2.3.5. Polyphenols.** The polyphenols were characterized by the reaction with ferric chloride (FeCl3). To 1 mL of methanolic extract, the addition of one drop of a 2% alcoholic ferric chloride solution resulting in a green coloration indicates the presence of polyphenols [19].

## 2.4. Total chemical quantification

**2.4.1. Polyphenols.** Spectrophotometric analysis was performed using the Folin-Ciocalteu reagent, following a modified method previously described by [20]. 20 µL of each wood tar extract is mixed with 1.16 ml distilled water, 100 µL of Folin-Ciocalteu reagent and 300 µL of freshly prepared 20% sodium carbonate ($Na_2CO_3$). Absorbance against a blank was determined at 765 nm after 30 minutes. The total phenolic content was expressed as micrograms of gallic acid equivalents per milligram of extract (µg GAE/mg).

**2.4.2. Flavonoids.** A modified method based on the proposed by [21] was used. 0.5 mL of the extract was mixed with 0.5 mL aluminum chloride ($AlCl_3$) and incubate at room temperature for 1 hour then its absorbance was measured against a blank at 420 nm. Total flavonoid content was expressed as micrograms of quercetin equivalents per milligram of extract (µg QE/mg).

**2.4.3. Tannins.** The Folin and Ciocalteu methods were used to create standard curves for tannin [22]. 1.5 mL of 20% sodium carbonate and 0.5 mL of Folin-phenol reagent were combined with 6.9 mL of distilled water and 0.1 mL of the extracts. The mixture was shaken well and allowed to sit at room temperature for 1 h. The absorbance was measured at 725 nm in a spectrophotometer. Tannic acid (µg TAE/mg) is used to express the results of tannins.

## 2.5. Antioxidant activity

**2.5.1. 2,2-Diphenyl-1-picrylhydrazylradical (DPPH) radical assay.** The DPPH radical scavenging activity of *JPWT* was determined according to the method described by Ouakil et al. [23]. Briefly, 2 mL of DPPH solution (0.02 mM in

methanol) was mixed with 50 µL of JPWT solution prepared in methanol. The mixture was incubated for 30 min in the dark at room temperature, and absorbance was measured at 517 nm. A control was prepared using 2 mL of DPPH solution and 50 µL of methanol. The percentage of DPPH inhibition was calculated, and results were expressed as $IC_{50}$ (µg/mL), defined as the concentration required to reduce the initial DPPH concentration by 50%. Tests were carried out in triplicate using ascorbic acid and butylated hydroxytoluene (BHT) as positive controls. Radical scavenging activity was expressed as percentage inhibition of DPPH radical and was calculated by following the equation:

$$\% \ Inhibition = \frac{(Absorbance \ of \ control - Absorbance \ of \ samples)}{Absorbance \ of \ control} * 100$$

The antioxidant activity of wood cedar wood tar extracts was expressed as $IC_{50}$, defined as the concentration of the test material required to cause a 50% decrease in initial DPPH concentration.

**2.5.2. Ferric-reducing antioxidant power (FRAP) assay.** The ferric reducing antioxidant power (FRAP) of *JPWT* was evaluated according to Yen and Chen [24], with minor modifications. A 0.1% (w/v) solution of *JPWT* was prepared. Briefly, 1 mL of JPWT solution was mixed with 2.5 mL of phosphate buffer (0.2 M, pH 6.6) and 2.5 mL of potassium ferricyanide solution (1% w/v). The mixture was incubated at 50°C for 20 min, then 2.5 mL of trichloroacetic acid (10% w/v) was added. After centrifugation, 2.5 mL of the supernatant was mixed with 2.5 mL of distilled water and 0.5 mL of ferric chloride solution (0.1% w/v). Absorbance was measured at 700 nm. BHT was used as a positive control. The IC50 (µg/mL) value was calculated. All measurements were performed in triplicate.

## 2.6. Disc diffusion assay

The antibacterial activity of *JPWT* was tested against both gram-positive and Gram-negative bacteria at 25 mg/mL and 50 mg/mL by the disc diffusion method described by Kalemba and Kunicka [17], with minor modifications. The following bacterial strains were obtained at the Research Technology Platforms of Mohamed VI University of Sciences and Health: *Staphylococcus aureus* (methicillin-resistant, MRSA *6538*), *Staphylococcus aureus* ATCC *6538*, *Enterococcus faecalis 29212CQ*, *Pseudomonas aeruginosa* ATCC 27853, *Escherichia coli* (ESBL-EC) 2897K, *Escherichia coli* (CREC) 2878K, and *Escherichia coli* ATCC 1225. Each inoculum was adjusted to $1.10^6$ CFU/mL (0.5 McFarland), and 0.5 mL of each suspension was spread onto nutrient agar plates. Sterile 6 mm paper discs were impregnated with 10 µL of *JPWT* prepared at 25 mg/mL or 50 mg/mL (diluted in 10% Tween 80) and placed on the inoculated plates. Gentamicin 10 µg/disc (CN) and imipenem 10 µg/disc (IPM) were used as positive controls. Plates were incubated at 37°C for 24 h, and inhibition zones were measured in millimeters. Each assay was performed in triplicate.

## 2.7. GCMS analysis

Weigh 10 mg of the *JPWT* extract and add 1 ml of MeOH. After vortexing and ultrasonic solubilization, the solution is filtered. 100 µl of the filtered solution is taken and dried with nitrogen, then 100 µl of BSTFA (N, O-bis(trimethylsilyl) trifluoroacetamide) is added, and the mixture is incubated at 70°C for 20 min. 2 µl is injected for GC-MS analysis. GC-MS analysis was performed using an Agilent 8890 gas chromatograph coupled to an Agilent 7000D mass spectrometer. Separation was achieved on an HP-5MS capillary column (30 m × 0.25 mm i.d., 0.25 µm film thickness). Helium was used as the carrier gas at a constant flow rate of 1.2 mL/min. The oven temperature was programmed from 60 °C to 300 °C. The mass spectrometer operated in electron impact (EI) mode at 70 eV, with a source temperature of 230 °C. Mass spectra were recorded in scan mode over the m/z range 30–500, with a solvent delay of 4 min.

## 2.8. Preparation of proteins and docking protocol in MOE

To evaluate protein-ligand interactions, the crystal structures 4HBT (CTX-M-15 β-lactamase) and 5M19 (PBP2a) were prepared in MOE (2024.06) by deleting non-essential molecules, adding missing hydrogens, and assigning appropriate

protonation states. The six major constituents hydroxychavicol, 2,3-dehydroferruginol, eugenol, 3-methylcatechol, palmitic acid, and pyrocatechol were converted to MOE-compatible format and briefly minimized to remove any steric clashes [25]. Because 4HBT has no co-crystallized ligand, docking was performed in blind mode (no explicit "box" parameters in MOE), using Triangle Matcher for placement, London dG for initial scoring, and GBVI/WSA dG for refinement; each ligand underwent 15 independent runs, and the top 5 poses per ligand were retained. For 5M19, the protocol was validated by redocking the co-crystallized inhibitor at the exact experimental coordinates (MOE does not require box dimensions; the site is defined by the co-crystal position), yielding a heavy atom RMSD of 0.60 Å, which supports the reliability of the subsequent docking runs applied to the six constituents.

### 2.9. Statistical analysis

All experiments were performed at least in triplicate (n = 3). Results are expressed as mean ± standard deviation (SD). Statistical differences between groups were assessed using one-way analysis of variance (ANOVA) followed by Tukey's multiple comparison test. Differences were considered statistically significant at $p < 0.05$.

## 3. Results

### 3.1. Preliminary phytochemical tests

Phytochemical screening was conducted to determine the presence (+) or absence (-) of principal secondary metabolites, namely, alkaloids, polyphenols, flavonoids, tannins, and saponins in *JTWT*. Qualitative determination of the *JTWT* extract determined the presence of multiple bioactive compounds, especially tannins, polyphenols, and flavonoids, but no saponins were present (Table 1).

### 3.2. Total chemical quantification

The quantification of secondary metabolites revealed that JPWT contained substantial amounts of tannins (110.59 ± 11.31 µg TAE/mg), polyphenols (242.91 ± 68.93 µg GAE/mg) and flavonoids (26.74 ± 0.52 µg QE/mg) (Table 2).

### 3.3. GCMS analysis

GC-MS analysis of *JPWT* led to the identification of 17 compounds (Table 4). The profile was dominated by phenolic and diterpenoid constituents, with hydroxychavicol (46.3%) and 2,3-dehydroferruginol (18.5%) as the most abundant

**Table 1. Screening of phytochemicals in *Juniperus phoenicea* wood tar.**

| Plant Extract | Tanin | Alkaloid | Polyphenols | Saponin | Flavonoid |
|---|---|---|---|---|---|
| *JPWT* | + | + | + | – | + |

+ indicates the presence of secondary metabolites in extracts, while – indicates the absence in qualitative screening phytochemical analysis.

**Table 2. Total phenolic, flavonoids, and Tannins contents in Juniperus phoenicea wood tar.**

| Plant extract | TTC (µg TAE/mg) | TPC (µg GAE/mg) | TFC (µg QE/mg) |
|---|---|---|---|
| *JPWT* | 110.59 ± 11.31[a] | 242.91 ± 68.93[b] | 26.74 ± 0.52[a] |

Data represents the mean ± standard deviation of three independent experiments. Values in the same row followed by different superscript letters (a-b) are significantly different according to Tukey's test ($p < 0.05$). A Tukey's test (α = 0.05) revealed no significant difference between TFC and TTC (p = 0.186). In contrast, TFC showed a significant difference compared to TTC (p = 0.047). A highly significant difference was observed between TPC and TFC (p = 0.004). These results indicate that TPC differs significantly from both TTC and TFC, whereas TFC and TTC present comparable mean values.

 

compounds, followed by eugenol (7.3%) and isoeugenol (3.9%). The predominance of these compounds may contribute to the observed antioxidant and antibacterial activities Table 3 and Fig 1.

### 3.4. Antioxidant activities

According to Table 3 findings, *JPWT* demonstrates significant antioxidant properties compared to the standard anti-oxidants BHT and Ascorbic acid, used as reference points ($p < 0.05$). The extract shows strong scavenging action on DPPH radicals and FRAP assays, with $IC_{50}$ values of $17.05 \pm 0.67$ and $75.81 \pm 2.02$ μg/mL, respectively. Nevertheless, this reducing power is still less active as of the synthetic antioxidants BHT ($IC_{50} = 24.02 \pm 2.03$ μg/mL) and Ascorbic acid ($IC_{50} = 1.996 \pm 0.1$ μg/mL). Furthermore, in the FRAP assay, BHT reduces iron with an $IC_{50}$ of $16.7 \pm 1.6$ μg/mL ($p < 0.05$) Table 3.

### 3.5. Antibacterial activity

*JPWT* exhibited antibacterial activity against the tested strains (Table 5). Inhibition zones increased with concentration, indicating dose-dependent effects. Overall, Gram-positive bacteria (*Staphylococcus aureus* and *Enterococcus faecalis*) were more susceptible than Gram-negative strains, which is consistent with the barrier effect of the outer membrane in Gram-negative bacteria. Among the Gram-positive bacteria, *Staphylococcus aureus* (methicillin-resistant, MRSA) and *Staphylococcus aureus* ATCC were the most susceptible to *JPWT*, with inhibition zones of $12.5 \pm 0.70$ mm and $13 \pm 0.0$ mm, respectively, at 50 mg/mL, compared to $8 \pm 1.41$ mm and $10 \pm 0.0$ mm at 25 mg/mL. *Enterococcus faecalis* exhibited moderate sensitivity, with inhibition zones measuring $6.5 \pm 0.70$ mm at 25 mg/mL and $11 \pm 1.41$ mm at 50 mg/mL. Compared to the reference antibiotic gentamicin (CN), which produced inhibition zones ranging from $10 \pm 0.0$ to $21 \pm 0.70$ mm, the extract showed lower but still notable activity, particularly against Staphylococcus species. For Gram-negative bacteria, JPWT had generally weaker inhibitory effects. *Pseudomonas aeruginosa* ATCC showed minimal sensitivity, maintaining an inhibition zone of $8 \pm 0.0$ mm at both concentrations, whereas imipenem (IPM), used as a

**Table 3. Chemical composition of *JPWT* as identified by GC-MS.**

| N° | Name of compounds | Retention time (min) | Area (%) |
|---|---|---|---|
| 1 | Phenol | 14.58 | 0.65 |
| 2 | 3-Methylphenol | 17.04 | 0.28 |
| 3 | 4-Methylphenol | 17.41 | 0.83 |
| 4 | 2-Methylphenol | 17.73 | 1.05 |
| 5 | Pyrocatechol | 24.12 | 14.21 |
| 6 | 3-Methylcatechol | 25.78 | 15.13 |
| 7 | 2,3-Dimethylhydroquinone | 27.04 | 7.74 |
| 8 | 3-Isopropylpyrocatechol | 28.24 | 2.81 |
| 9 | Eugenol | 28.61 | 11.37 |
| 10 | Hydroxychavicol | 29.63 | 1.92 |
| 11 | 3-Vanilpropanol | 31.64 | 3.73 |
| 12 | Palmitic Acid | 33.82 | 5.36 |
| 13 | 2,3-Dehydroferruginol | 35.75 | 6.45 |
| 14 | Pisiferal | 36.55 | 13.21 |
| 15 | Homovanillyl alcohol | 37.24 | 3.65 |
| 16 | Ethisterone, trimethylsilyl ether, O-methyloxime | 38.83 | 7.10 |
| 17 | 4,4'-Methylenebis [2-nitrobenzenamine] | 41.16 | 4.44 |

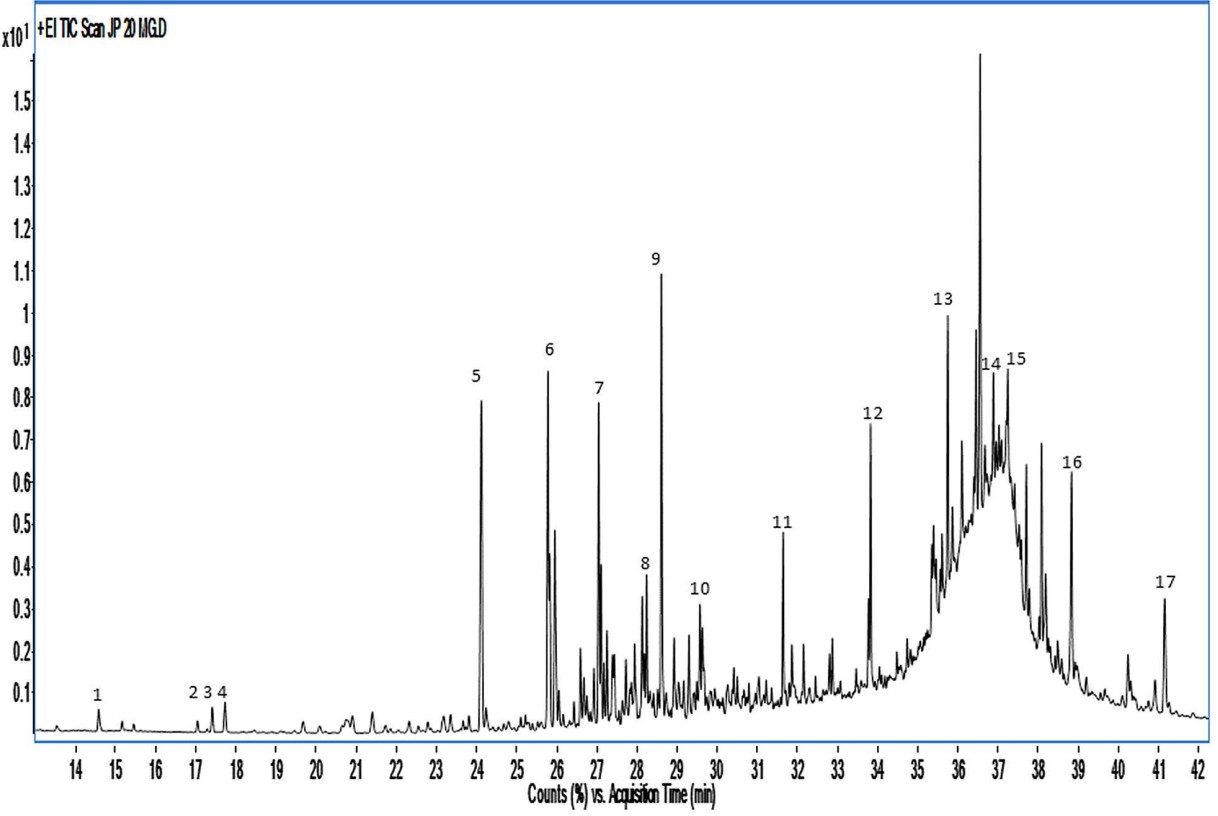

**Fig 1. GC-MS of *JPWT* showing the identified phytochemical constituents.**

positive control, produced a zone of 23 ± 0.0 mm. Conversely, *Escherichia coli* (ESBL-producing) showed moderate susceptibility to *JPWT*, with inhibition zones increasing from 6 ± 0.0 mm at 25 mg/mL to 12.5 ± 0.70 mm at 50 mg/mL, indicating a dose-dependent effect. The extract also had limited activity against carbapenem-resistant *E. coli* and *E. coli* ATCC, with inhibition zones ranging from 6 ± 0.0 to 9 ± 0.0 mm, while positive controls imipenem and gentamicin produced inhibition zones varying from 0 ± 0.0 to 25 ± 0.0 mm depending on the strain Table 5.

### 3.6. Molecular docking

Table 6 shows that two hydrophobic scaffolds dominate across both targets. For PBP2a (5M19), 2,3-dehydroferruginol (−7.04 kcal·mol⁻¹) and palmitic acid (−6.45 kcal·mol⁻¹) yield the most favorable scores, each surpassing the re-docked co-crystal reference (−5.50 kcal·mol⁻¹; 0.60 Å RMSD), which indicates pose fidelity and suggests these phytochemicals can compete effectively at the PBP2a binding site. The smaller phenolics eugenol, hydroxychavicol, 3-methylcatechol, and pyrocatechol (≈ −4.2 to −4.8 kcal·mol⁻¹) score consistently weaker, consistent with limited hydrophobic surface complementarity needed to stabilize interactions in PBP2a. For CTX-M-15 (4HBT) docked in blind mode due to the apo structure palmitic acid again ranks first (−6.01 kcal·mol⁻¹) followed by 2,3-dehydroferruginol (−5.34 kcal·mol⁻¹), while the remaining phenolics cluster around −4.0 to −4.6 kcal·mol⁻¹, indicating modest affinity. The cross-target convergence therefore nominates palmitic acid and 2,3-dehydroferruginol as priority leads for follow-up against both MRSA PBP2a and ESBL CTX-M-15 (Figs 2, 3).

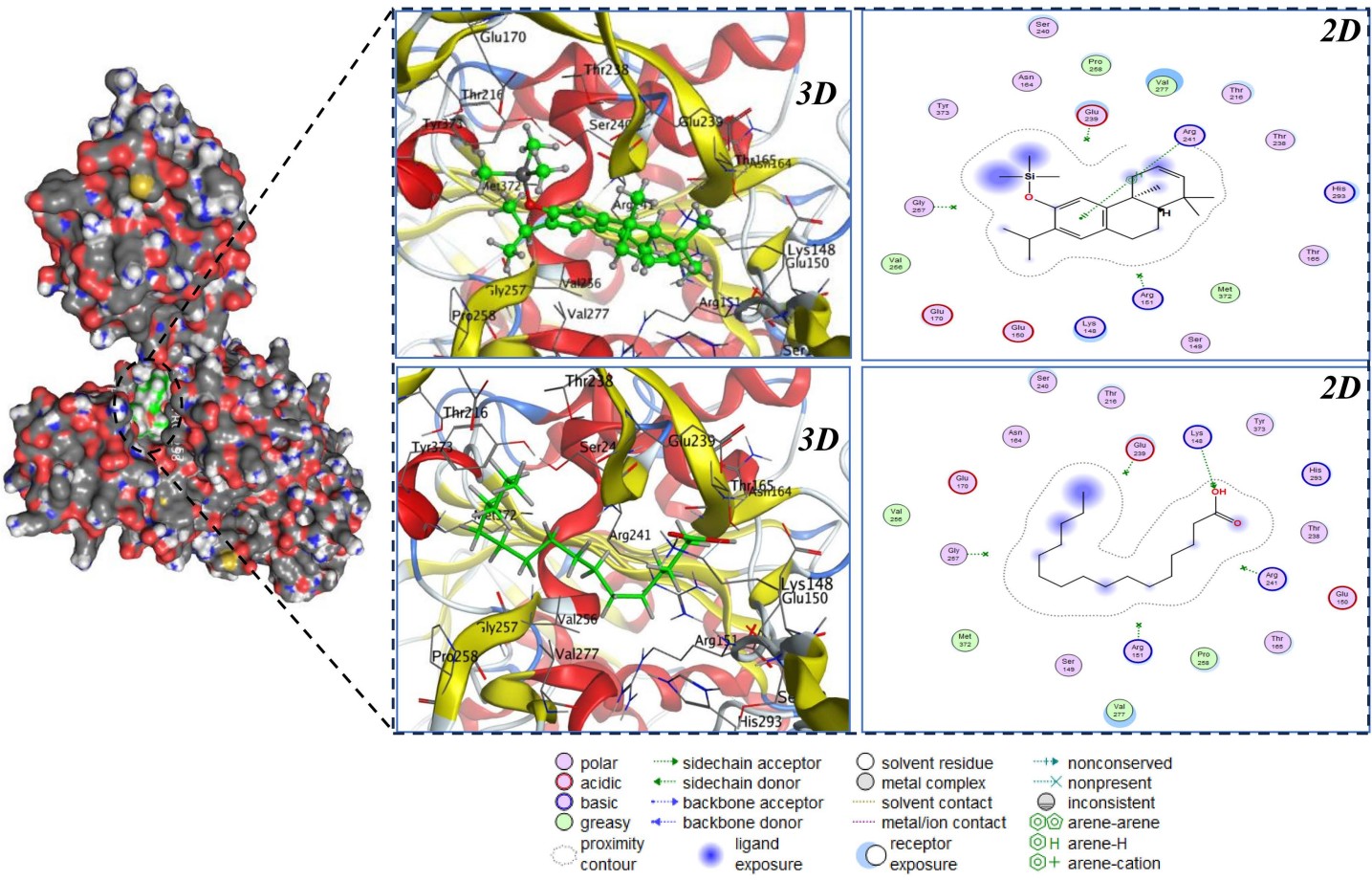

**Fig 2. 2D/3D binding poses and interaction fingerprints of palmitic acid and 2,3-dehydroferruginol in PBP2a (*PDB* 5M19), showing convergence to the same lipophilic cleft.**

## 4. Discussion

*JPWT* showed a broad qualitative phytochemical profile, with detectable polyphenols, tannins, flavonoids and alkaloids (Table 1) [26]. The high levels of total polyphenols and tannins (Table 2) are consistent with the radical-scavenging and reducing capacities observed in DPPH and FRAP assays (Table 3). Antioxidants protect biological systems against oxidative stress by scavenging free radicals and preventing lipid peroxidation. Currently, plant-derived natural antioxidants are increasingly recommended for the prevention and treatment of various diseases [27]. The antioxidant activity of *JPWT* was evaluated using the DPPH assay, a widely accepted method for assessing free radical scavenging capacity [28]. *JPWT* exhibited strong antioxidant activity with an $IC_{50}$ value of $17.05 \pm 0.67$ µg/mL, which was higher than BHT ($IC_{50} = 24.02 \pm 2.03$ µg/mL) but lower than ascorbic acid ($IC_{50} = 1.996 \pm 0.1$ µg/mL), in agreement with previous studies [26]. The FRAP assay, a rapid and cost-effective method for evaluating total antioxidant capacity [29], showed that BHT had the highest ferric reducing power, whereas *JPWT* displayed a lower reducing capacity ($75.81 \pm 2.02$ µg/mL). This activity is likely related to the presence of secondary metabolites (Table 4), particularly phenolic constituents such as hydroxychavicol and eugenol, which are known to act as hydrogen/electron donors and can therefore contribute to antioxidant activity [30–32]. However, Tannins, and flavonoids compounds can contribute to reducing oxidative stress and lowering the risk of metabolic and degenerative disorders [33–38].

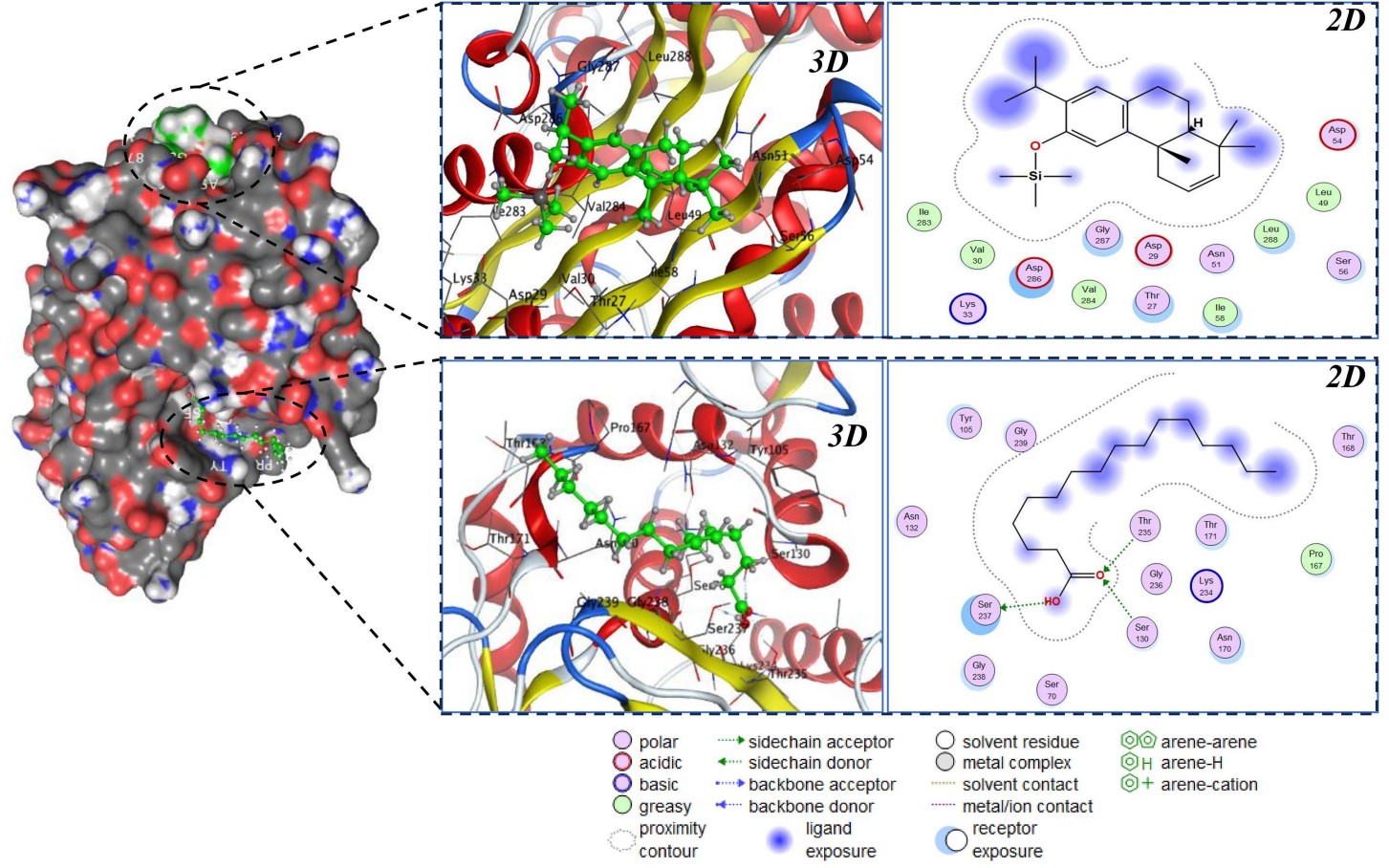

**Fig 3. 2D/3D binding poses and interaction fingerprints of palmitic acid and 2,3-dehydroferruginol in CTX-M-15β-lactamase (*PDB* 4HBT), showing occupation of distinct micro-sites.**

**Table 4. Antioxidant activity of JPWT (IC50 µg/mL; mean ± SD, n = 3).**

| Assays | JPWT | Positive control | |
|---|---|---|---|
| | | BHT | Ascorbic acid |
| DPPH | 17.05 ± 0.67[a] | 24.02 ± 2.03[a] | 1.996 ± 0.1[a] |
| FRAP | 75.81 ± 2.02[a] | 16.7 ± 1.6[b] | --- |

$IC_{50}$ = Half maximal inhibitory concentration. Different superscript letters within the same row indicate significant differences according to Tukey's test ($\alpha = 0.05$). The results are expressed as mean values and standard deviation (n = 3). Tukey's test ($\alpha = 0.05$) for the DPPH radical scavenging assay showed no significant difference between BHT and *JPWT* (p = 0.341) nor between Ascorbic acid and *JPWT* (p = 0.070). In contrast, an important difference was observed between Ascorbic acid and BHT (p = 0.025). While no significant differences were observed between *JPWT* and the reference antioxidants in the DPPH assay (p > 0.05), a different trend was noted in the FRAP assay. Tukey's test revealed a significant difference between *JPWT* and BHT in FRAP (p = 0.001). Overall, these results indicate that *JPWT* exhibits DPPH radical scavenging activity comparable to both reference antioxidants, while showing a superior reducing capacity compared to the synthetic antioxidant (BHT), and a clear difference was observed between BHT and ascorbic acid.

The GC–MS analysis of *JPWT* (Table 3) revealed a complex phytochemical composition mainly characterized by phenolic derivatives and diterpenoid compounds. The most abundant constituents were 3-methylcatechol (15.13%), pyrocatechol (14.21%). Several bioactive constituents were identified, including phenol and its methylated derivatives (2-, 3-, and 4-methylphenol), catechol derivatives (pyrocatechol, 3-methylcatechol, 2,3-dimethylhydroquinone, and 3-isopropylpyrocatechol), as well as eugenol, homovanillyl alcohol, hydroxychavicol, palmitic acid, 3-vanilpropanol, 2,3-dehydroferruginol, pisiferal, ethisterone trimethylsilyl ether O-methyloxime, and 4,4'-methylenebis [2-nitrobenzenamine] [39]. These compounds are widely reported to exhibit antimicrobial, antioxidant, anti-inflammatory, and anticancer activities, while palmitic acid has been associated with lipid metabolism and cardiovascular effects [31,40–49]. Similar phenolic and terpenoid constituents have been reported in *Juniperus phoenicea* derived preparations and are frequently linked to antimicrobial and antioxidant activities [50,51]. Although essential oils and wood tars differ in extraction processes and chemical composition, the presence of these bioactive compounds supports the pharmacological potential of *JPWT* and corroborates the biological activities observed in the present study.

Over the past three decades, pharmaceutical companies have created a variety of innovative antibiotic treatments, but bacteria have grown more resistant to these medications. Plant extracts are a fantastic source of pathogen-fighting antibacterial compounds [52]. They can therefore be utilized to treat a variety of infectious disorders brought on by virulent microorganisms. In the antibacterial screening, *JPWT* was more active against Gram-positive strains than Gram-negative bacteria (Table 5). This trend is commonly attributed to the presence of an outer membrane in Gram-negative bacteria, which restricts the penetration of hydrophobic compounds. Phenolic compounds can disrupt bacterial membranes and interfere with enzyme systems, which may explain the inhibitory effects observed for *JPWT*. Additionally, there are not many reports on the antibacterial properties of *JPWT*. The results of the current investigation make it abundantly evident that *JPWT* proven to have significant antibacterial activity.

Structure-based molecular docking offers a mechanistic bridge between our disc-diffusion phenotypes and atomistic recognition at resistance-defining enzymes, enabling us to rationalize activity and prioritize leads before costly assays. Guided by the highest inhibition zones observed against ESBL *E. coli* and MRSA, we focused on two clinically pivotal targets: CTX-M-15, a class A extended-spectrum β-lactamase (PDB 4HBT) that hydrolyzes third-generation cephalosporins and undermines β-lactam efficacy in Gram-negative pathogens, and PBP2a, the low-affinity penicillin-binding protein (PDB 5M19) that preserves transpeptidation and cell-wall cross-linking in MRSA despite β-lactam exposure [53]. Docking these enzymes against our six major constituents hydroxychavicol, 2,3-dehydroferruginol, eugenol, 3-methylcatechol,

**Table 5. Inhibition zone diameter (mm) (mean±SD, n=3) of JPWT against selected bacteria and positive controls.**

| Bacterial strain | Inhibition zone (mm) | | Positive control (Antibiotic) |
|---|---|---|---|
| | 25 mg/mL | 50 mg/mL | |
| *Staphylococcus aureus* (methicillin-resistant, MRSA) | 8±1.41[a] | 12.5±0.70[b] | (CN) − 21±0.70[c] |
| *Staphylococcus aureus* ATCC | 10±0[a] | 13±0[b] | (CN) − 19±0[c] |
| *Enterococcus faecalis* | 6.5±0.70[a] | 11±1.41[a] | (CN) − 10±0[a] |
| *Pseudomonas aeruginosa* ATCC | 8±0[a] | 8±0[a] | (IPM) − 23±0[b] |
| *Escherichia coli* (ESBL-producing) | 6±0[a] | 12.5±0.70[b] | (IPM) − 22±1.41[c] |
| *Escherichia coli* (carbapenem-resistant) | 7±0[a] | 7±0[a] | (CN) − 0±0[a] |
| *Escherichia coli* ATCC | 6±0[a] | 9±0[b] | (IPM) − 25±0[c] |

Data represent the mean±standard deviation of three independent experiments. Different superscript letters within the same row indicate significant differences according to Tukey's post hoc test (α=0.05). Tukey's post hoc test (α=0.05) revealed a significant difference between the concentrations of 50 mg/mL and 25 mg/mL for most tested strains, except for *Pseudomonas aeruginosa*. Comparisons with reference antibiotics showed significant differences between both concentrations and Gentamicin for MRSA and *Staphylococcus aureus* ATCC, while no significant differences were observed for *Enterococcus faecalis*. For Gram-negative bacteria, Imipenem differed significantly from 25 mg/mL for *P. aeruginosa* and ESBL-producing strains, whereas a significant difference with 50 mg/mL was observed only for ESBL strains.

palmitic acid, and pyrocatechol probes complementary interaction chemistries: phenolic donors/acceptors to engage catalytic serine-centered machinery and polar subsites, and hydrophobic/aryl surfaces to occupy lipophilic grooves and π-stacking regions that modulate catalysis or access channels. By estimating relative binding affinities and preferred poses, docking tests the plausibility that these phytochemicals can attenuate β-lactamase turnover or disrupt PBP-mediated cross-linking, thereby offering a structure-guided rationale for the observed antibacterial trends across Gram-negative and Gram-positive backgrounds.

Fig 2 (2D/3D) depicts the interaction fingerprints, and the 3D pose overlays of 2,3-dehydroferruginol and palmitic acid in PBP2a (5M19), showing that both ligands occupy essentially the same subsite within a continuous lipophilic cleft bounded on the polar edge by Lys148-Glu239-Arg241 and along the hydrophobic wall by Val256/Val277-Pro258-Met372. In 2D, 2,3-dehydroferruginol presents its phenolic/aromatic head toward Arg241, consistent with a side-chain H-bond/cation-π anchor, while its fused hydrocarbon core packs against Val277/Pro258 with additional van der Waals contacts to Gly257/Met372 [54,55]; the 3D view corroborates deep insertion and tight contour complementarity throughout this groove. Palmitic acid follows the same trajectory: the long aliphatic chain traces the Val-Pro-Met hydrophobic track, whereas the carboxyl headgroup orients toward Lys148 (side-chain donor H-bond) with a secondary polar engagement to Arg241; in 3D, the head sits at the polar rim while the tail fully buries along the cleft, consistent with the 2D exposure halos. The pose convergence of these chemically distinct ligands in the same pocket explains their similarly favorable docking energies in Table 1, with 2,3-dehydroferruginol slightly outperforming palmitic acid due to additional π-driven stabilization around Arg241 and superior shape complementarity evident in both the 2D contacts and the 3D overlays.

Based on the 2D/3D visualization in Fig 3, both ligands bind within the CTX-M-15 β-lactamase (4HBT) pocket but occupy distinct micro-sites that explain their energy ranking in Table 6. 2,3-Dehydroferruginol settles on a predominantly hydrophobic shelf flanked by Val30/Val284/Leu288/Ile283-Ile58 and bordered peripherally by Asp29/Asp286/Asp54, with only weak polar opportunities nearby [55–57]; its aromatic-alicyclic framework is stabilized mainly by dispersion/van der Waals contacts and shows noticeable ligand exposure in the 2D map, consistent with a shallower, entrance-proximal pose and a moderate score ($-5.34$ kcal·mol$^{-1}$). In contrast, palmitic acid threads along an elongated tunnel that traverses the catalytic rim engaging the Ser130 region and the β3-β4 segment (Lys234-Thr235-Ser237-Gly238/Gly239) while running parallel to the Ω-loop residues (Thr168/Thr171/Asn170). The carboxyl headgroup forms a small H-bond network (Ser237/Thr235, with possible electrostatic assistance from Lys234), whereas the aliphatic chain is deeply buried against the lipophilic track, producing minimal exposure across the chain in 2D and a snug, continuous burial in 3D. This combination of specific polar anchoring near the catalytic machinery and extensive hydrophobic complementarity rationalizes the more favorable docking energy for palmitic acid ($-6.01$ kcal·mol$^{-1}$) relative to 2,3-dehydroferruginol, and supports the

**Table 6. MOE docking energies (kcal·mol$^{-1}$) of six major constituents against PBP2a (5M19) and CTX-M-15 β-lactamase (4HBT), with co-crystal redocking control for 5M19.**

| Binding Energy (Kcal/mol) | | |
|---|---|---|
| PDB code/compounds | 5M19 | 4HBT |
| Hydroxychavicol | −4.71 | −4.46 |
| 2,3-Dehydroferruginol | −7.04 | −5.34 |
| Eugenol | −4.79 | −4.58 |
| 3-Methylcatechol | −4.17 | −4.13 |
| Palmitic Acid | −6.45 | −6.01 |
| Pyrocatechol | −4.19 | −3.96 |
| Co-5M19 | −5.50 | *** |
| 4HBT | *** | *** |

hypothesis that palmitate-like scaffolds can more effectively occupy the active-site corridor of CTX-M-15 and potentially impede access/turnover of β-lactam substrates [58,59].

Overall, the docking analysis (Table 6; Figs 2, 3) identifies palmitic acid and 2,3-dehydroferruginol as the most promising binders across both targets. On PBP2a, the two ligands converge into the same lipophilic pocket with extensive nonpolar burial, whereas on CTX-M-15 palmitic acid penetrates deeper into the active-site corridor with more balanced polar/hydrophobic complementarity explaining its superior score. The workflow is supported by sub-ångström redocking accuracy on PBP2a (RMSD = 0.60 Å), whereas the apo CTX-M-15 poses require further structural refinement and conformational optimization before proceeding to biochemical validation.

Our findings suggest that *JPWT* may serve as a natural antioxidant capable of mitigating oxidative stress. However, further studies are required to isolate and purify the bioactive compounds responsible for their antioxidant and antibacterial activities. From an applied perspective, wood tars obtained from conifers are traditionally produced and used in Morocco [60,61], but their production can raise sustainability concerns if harvesting is not controlled. Ethnobotanical investigations have highlighted the socio-economic importance of medicinal tars as well as the potential pressure on local biodiversity [60,62]. The valorization of *JPWT* as a source of natural antimicrobial and antioxidant agents should therefore be coupled with sustainable collection practices. In addition, because GC-MS mainly detects volatile/semi-volatile constituents (often after derivatization), complementary techniques such as LC-MS/MS are recommended in future work to better characterize non-volatile polar metabolites.

## 5. Conclusions

This study establishes that *JPWT* exhibits significant antioxidant and antimicrobial activities, largely attributable to its abundant phenolic, tannin, and flavonoid constituents. GC-MS profiling uncovered a diverse chemical composition predominantly featuring hydroxychavicol, eugenol, and 2,3-dehydroferruginol compounds recognized for their established biological properties. Molecular docking studies corroborated antimicrobial experimental findings, revealing strong binding interactions between palmitic acid and 2,3-dehydroferruginol with critical bacterial targets (PBP2a and CTX-M-15 β-lactamase), thereby offering mechanistic insights into the observed antimicrobial effects. These results position *JPWT* as a promising natural product for further development. However, additional studies are required to isolate active compounds, determine MIC/MBC values, assess safety, and evaluate efficacy in relevant in vivo models. Sustainable valorization of traditional wood tar resources could also contribute to local socio-economic development when coupled with conservation-oriented practices

## Acknowledgments

This work was supported and funded by the Deanship of Scientific Research at Imam Mohammad Ibn Saud Islamic University (IMSIU) (grant number IMSIU-DDRSP2601). The authors gratefully acknowledge the HPC Marwan team for providing access to high-performance computing resources used to perform the molecular docking calculations. We also thank all colleagues who contributed to sample collection and laboratory assistance.

## Author contributions

**Conceptualization:** Sadia Tina, Oussama Khibech, Samy Iskandar, Allal Challioui, Balouch Lhousaine, Meryem El Jemli.

**Data curation:** Sadia Tina.

**Formal analysis:** Sadia Tina, Oussama Khibech, Mohammed Al-zharani, Fahd A. Nasr.

**Funding acquisition:** Meryem El Jemli.

**Investigation:** Oussama Khibech.

**Methodology:** Sadia Tina, Oussama Khibech.

**Resources:** Balouch Lhousaine.

**Software:** Sadia Tina, Oussama Khibech.

**Validation:** Mohamed Bouhrim, Meryem El Jemli.

**Writing – original draft:** Sadia Tina, Oussama Khibech, Ashraf Ahmed Qurtam.

**Writing – review & editing:** Samy Iskandar, Meryem El Jemli.

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
