## [Decision Letter · Decision Letter 0]

4 Jan 2026

Dear Dr. Tina,

Thank you for submitting your manuscript to PLOS ONE. After careful consideration, we feel that it has merit but does not fully meet PLOS ONE’s publication criteria as it currently stands. Therefore, we invite you to submit a revised version of the manuscript that addresses the points raised during the review process.

We look forward to receiving your revised manuscript.

Kind regards,

Marcello Iriti, Ph.D.

Academic Editor

PLOS One

**Journal Requirements:**

1. When submitting your revision, we need you to address these additional requirements. Please ensure that your manuscript meets PLOS ONE's style requirements, including those for file naming. The PLOS ONE style templates can be found at https://journals.plos.org/plosone/s/file?id=wjVg/PLOSOne_formatting_sample_main_body.pdf and https://journals.plos.org/plosone/s/file?id=ba62/PLOSOne_formatting_sample_title_authors_affiliations.pdf 2. We noticed you have some minor occurrence of overlapping text with the following previous publication(s), which needs to be addressed: GCMS-based phytochemical profiling and in vitro pharmacological activities of plant Alangium salviifolium (L.f) Wang - https://doi.org/10.1186/s43094-024-00631-3 Phytochemical composition and in vitro antioxidant and antimicrobial activities of Bersama abyssinica F. seed extracts - https://doi.org/10.1038/s41598-024-56659-1 In your revision ensure you cite all your sources (including your own works), and quote or rephrase any duplicated text outside the methods section. Further consideration is dependent on these concerns being addressed. 3. In your Methods section, please provide additional information regarding the permits you obtained for the work. Please ensure you have included the full name of the authority that approved the field site access and, if no permits were required, a brief statement explaining why. 4. Thank you for stating the following financial disclosure: Biology Department, College of Science, Imam Mohammad Ibn Saud Islamic University (IMSIU) (grant number IMSIU-DDRSP2501), Riyadh, 11623, Saudi Arabia   Please state what role the funders took in the study.  If the funders had no role, please state: "The funders had no role in study design, data collection and analysis, decision to publish, or preparation of the manuscript." If this statement is not correct you must amend it as needed. Please include this amended Role of Funder statement in your cover letter; we will change the online submission form on your behalf. 5. Thank you for stating the following in the Acknowledgments Section of your manuscript: This work was supported and funded by the Deanship of Scientific Research at Imam Mohammad Ibn Saud Islamic University (IMSIU) (grant number IM-SIU-DDRSP2501). We gratefully acknowledge the HPC Marwan team for granting us privileged access to their high-performance computing resources. The exceptional computational power and responsive technical support they provided were instrumental in carrying out the molecular-dynamics simulations reported in this study. Their contribution greatly enhanced the quality and robustness of our results, and we are sincerely thankful for their assistance. We note that you have provided funding information that is not currently declared in your Funding Statement. However, funding information should not appear in the Acknowledgments section or other areas of your manuscript. We will only publish funding information present in the Funding Statement section of the online submission form. Please remove any funding-related text from the manuscript and let us know how you would like to update your Funding Statement. Currently, your Funding Statement reads as follows: Biology Department, College of Science, Imam Mohammad Ibn Saud Islamic University (IMSIU) (grant number IMSIU-DDRSP2501), Riyadh, 11623, Saudi Arabia  Please include your amended statements within your cover letter; we will change the online submission form on your behalf. 6. Please upload a new copy of Figures 1 and 2 as the detail is not clear. Please follow the link for more information:  https://journals.plos.org/plosone/s/figures 7. If the reviewer comments include a recommendation to cite specific previously published works, please review and evaluate these publications to determine whether they are relevant and should be cited. There is no requirement to cite these works unless the editor has indicated otherwise. 

Reviewers' comments:

**Comments to the Author**

1. Is the manuscript technically sound, and do the data support the conclusions?

Reviewer #1: Yes

Reviewer #2: Yes

2. Has the statistical analysis been performed appropriately and rigorously?

Reviewer #1: No

Reviewer #2: No

3. Have the authors made all data underlying the findings in their manuscript fully available?

Reviewer #1: Yes

Reviewer #2: Yes

4. Is the manuscript presented in an intelligible fashion and written in standard English?

Reviewer #1: No

Reviewer #2: No

**Reviewer #1:** Manuscript: “Chemical characterization, antimicrobial, and antioxidant potentials of Juniperus phoeniceae wood tar”. Manuscript: “Chemical characterization, antimicrobial, and antioxidant potentials of Juniperus phoeniceae wood tar”. Manuscript: “Chemical characterization, antimicrobial, and antioxidant potentials of Juniperus phoeniceae wood tar”. Manuscript: “Chemical characterization, antimicrobial, and antioxidant potentials of Juniperus phoeniceae wood tar”.

The manuscript provides general information on the antibacterial and antioxidant activities of Juniperus phoenicea wood tar (JPWT), supported by qualitative and quantitative phytochemical analyses. However, it requires thorough English language revision and addressing of the following major issues:

Title

Correct "Juniperus phoeniceae" to “Juniperus phoenicea”. Please go throughout the entire manuscript.

Abstract

Correct "Juniperus phoeniceae" to “Juniperus phoenicea L.” in the first citation.

Rewrite "confirming its free radical-scavenging capability" to "confirming its antioxidant activity." Note that "radical-scavenging capability" is appropriate only when using radicals such as DPPH.

Materials and Methods

Rewrite the sentence "Voucher specimen of JP is deposited in the Herbarium (RAB114890) of the Scientific Institute of Rabat (Morocco) (Branches were examined for integrity and absence of dust and insect contamination)" for clarity and completeness.

Rename section "2.3. Preliminary phytochemical tests" to "2.3. Qualitative phytochemical tests."

Section 2.3.1: Avoid starting sentences with numbers.

Rewrite "After adding 2 mL of distilled water was added to 2 mL ethanolic solution of JPWT" for grammatical accuracy.

Correct "2.4.2. Flavonoid" to "2.4.2. Flavonoids."

Correct "2.4.3. Tannin" to "2.4.3. Tannins."

Correct "2.5.1. 2,2-Diphenyl-1-picrylhydrazylradical assay (DPPH)" to "2.5.1. 2,2-Diphenyl-1-picrylhydrazyl (DPPH) radical assay."

Correct "Tests were carried out in triplicate using ascorbic acid and BHT has positive control" to "Tests were carried out in triplicate using ascorbic acid and BHT as positive controls."

Use consistent nomenclature throughout (e.g., J. phoeniceae or JPWT instead of "wood cedar" in section 2.5.1).

Specify the tested microorganisms in the sentence: "A suspension of the tested microorganism (0.5 mL of 106 cells/mL) was spread on nutrient agar."

Define abbreviations in the sentence: "Positive controls were prepared using the CN and IPM antibiotics."

Section 2.7. GC-MS analysis: Clarify how phytocompounds were identified.

Consider switching to HPLC-MS, which is more suitable for polyphenol, alkaloid, and flavonoid-rich extracts.

Define BSTFA.

Results

Correct all "Tableau X" to "Table X" (e.g., Table 1, Table 2… Table 6).

Use decimal points (.) consistently in values for Tables 2 and 3.

In Table 3, specify units for IC50

Section 3.4. GC-MS analysis: Avoid detailing retention times (RT) of identified compounds.

Correct "3.5. Antibacterial stains" to "3.5. Antibacterial activity."

Please insert the sentence "The antibacterial activity of JPWT was tested against both Gram-positive and Gram-negative bacteria at concentrations of 25 mg/mL and 50 mg/mL" in the Materials and Methods section.

Italicize scientific names (e.g., Staphylococcus aureus).

Discussion

Remove "Firstly,".

Rewrite "Phytochemical screening revealed the presence of alkaloids, tannins, phenols, and flavonoids (Table 1). Quantification of total phenolics, tannins, and flavonoids (Table 2) by UV-Visible spectroscopy demonstrated that JPWT is rich in these compound", to include in-depth literature-based discussion specific to J. phoeniceae.

Rewrite Table 5 title to "Inhibition zone diameter (mm) of Juniperus phoenicea wood tar and positive controls (antibiotics) against selected bacteria."

Report values in Table 5 as means of three replicates ± standard deviation, with appropriate statistical analysis.

Conclusions

Correct "Collectively, these results position JPWT as a valuable reservoir of bioactive natural products with considerable pharmaceutical and therapeutic promise" for grammatical accuracy and precision.

References

Complete the reference : El Jemli M. "Contribution to the ethnobotanical, toxicological, pharmacological and phytochemical study of four Moroccan Cupressaceae: Juniperus thurifera L., Juniperus oxycedrus L., Juniperus phoenicea L. and Tetraclinis articulata L." (Provide full details: thesis/degree, institution, year, etc., 2020).

Please recheck the whole reference list.

**Reviewer #2:** Submission ID: PONE-D-25-64294 Submission ID: PONE-D-25-64294 Submission ID: PONE-D-25-64294 Submission ID: PONE-D-25-64294

Journal: PLOS ONE

Dear authors,

Your manuscript entitled “Chemical Characterization, antimicrobial, and antioxidant potentials of Juniperus phoeniceae wood tar” has been reviewed. The topic of your manuscript is very interesting, you are following the right scientific path for writing your manuscript, and the manuscript is well organised and presents important information for the scientific field. However, the manuscript needs some improvements. Therefore, I would like to make some comments, recommendations about the revised manuscript:

• The objective of your research should be discussed in detail in the last paragraph of the introduction.

• What is the importance of adding a Molecular Docking section? Normally, the in-silico study should be carried out before starting the study.

• What is the added value of Molecular Docking for your study?

• The references cited in the Molecular Docking section are not compatible with the contents. Please check these references.

• Add a section for statistical analysis and add the results of the statistical analysis.

• The Discussion section requires a recent reference in the relevant fields and should show the socio-economic importance of your results in relation to those in the literature.

• For the conclusion, present the most important results, the socio-economic importance of the results obtained, as well as recommendations and prospects for your research.

.

Reviewer #1: No

Reviewer #2: **Yes:** Azeddin EL BARNOSSIAzeddin EL BARNOSSIAzeddin EL BARNOSSIAzeddin EL BARNOSSI

---

## [Author Response · Author response to Decision Letter 1]

21 Jan 2026

The authors sincerely thank the editor and the reviewers for their valuable time, consideration, and suggestions.

---

## [Decision Letter · Decision Letter 1]

9 Feb 2026

Dear Dr. Tina,

Thank you for submitting your manuscript to PLOS ONE. After careful consideration, we feel that it has merit but does not fully meet PLOS ONE’s publication criteria as it currently stands. Therefore, we invite you to submit a revised version of the manuscript that addresses the points raised during the review process.

We look forward to receiving your revised manuscript.

Kind regards,

Marcello Iriti, Ph.D.

Academic Editor

PLOS One

Journal Requirements:

Reviewers' comments:

Reviewer's Responses to Questions

**Comments to the Author**

Reviewer #1: (No Response)

Reviewer #2: All comments have been addressed

2. Is the manuscript technically sound, and do the data support the conclusions?

Reviewer #1: Partly

Reviewer #2: Yes

3. Has the statistical analysis been performed appropriately and rigorously?

Reviewer #1: Yes

Reviewer #2: Yes

4. Have the authors made all data underlying the findings in their manuscript fully available?

Reviewer #1: No

Reviewer #2: Yes

5. Is the manuscript presented in an intelligible fashion and written in standard English?

Reviewer #1: No

Reviewer #2: Yes

Reviewer #1: The authors have addressed most of the previous comments, but some points require further revision and clarification. Thorough English language revision is still needed throughout the manuscript.

Reference(s) is needed for this statement: "Among its derivatives, Juniperus phoeniceae wood tar (JPWT), obtained by the destructive distillation of JP wood, is traditionally used for skin ailments and infections."

Correct "Juniperus phoeniceae” in the whole document.

Revise references for qualitative phytochemical tests, ensuring the use of relevant references and correct numbering/formatting according to journal instructions.

Specify the amount of each antibiotic/disc.

State exactly the number of replicates in the Statistical analysis section.

Avoid starting sentences with numbers.

Provide the source/references for the MRSA, ESBL-EC, and CREC strains used in this study, and complete the references for ATCC strains as well.

Revise the chemical composition table, as the percentage areas sum to more than 100%. Clarify how these percentages were calculated and how identification was performed. Add the chromatogram if applicable.

Please include the statistical significance in the tables 3 & 5, rather than in the footnote for greater clarity.

Revise the sequence and order of headings (eg. chemical composition, antioxidant, antibacterial sections) to ensure a logical flow.

Reviewer #2: Manuscript Number: PONE-D-25-64294R1

Journal: PLOS One

Dear authors,

With these significant revisions, I consider the article suitable for publication.

.

Reviewer #1: No

Reviewer #2: **Yes:** Azeddin EL BARNOSSIAzeddin EL BARNOSSIAzeddin EL BARNOSSIAzeddin EL BARNOSSI

---

## [Author Response · Author response to Decision Letter 2]

11 Mar 2026

The authors sincerely thank the editor and the reviewers for their valuable time, consideration, and suggestions. The constructive comments have significantly enhanced the overall quality of the manuscript. We trust that we have adequately addressed your inquiries and concerns

---

## [Decision Letter · Decision Letter 2]

13 Mar 2026

Chemical Characterization, antimicrobial, and antioxidant potentials of Juniperus phoenicea wood tar

PONE-D-25-64294R2

Dear Dr. Tina,

We’re pleased to inform you that your manuscript has been judged scientifically suitable for publication and will be formally accepted for publication once it meets all outstanding technical requirements.

Kind regards,

Marcello Iriti, Ph.D.

Academic Editor

PLOS One

Additional Editor Comments (optional):

Reviewers' comments:

Reviewer's Responses to Questions

**Comments to the Author**

Reviewer #1: All comments have been addressed

2. Is the manuscript technically sound, and do the data support the conclusions?

Reviewer #1: Yes

3. Has the statistical analysis been performed appropriately and rigorously?

Reviewer #1: Yes

4. Have the authors made all data underlying the findings in their manuscript fully available?

Reviewer #1: Yes

5. Is the manuscript presented in an intelligible fashion and written in standard English?

Reviewer #1: Yes

Reviewer #1: Dear authors,

The manuscript has been improved and is worthy of acceptance for publication.

.

Reviewer #1: No

---

## [Editor Report · Acceptance letter]

PONE-D-25-64294R2

PLOS One

Dear Dr. Tina,

I'm pleased to inform you that your manuscript has been deemed suitable for publication in PLOS One. Congratulations! Your manuscript is now being handed over to our production team.

Kind regards,

on behalf of

Prof. Marcello Iriti

Academic Editor

PLOS One